# Advancing Transdermal Delivery by Zn/Ag-Electrode-Printed Iontophoretic Patch with Self-Generating Microcurrents

**Cheng-Liang Peng** [1,†] [ID], **Pei-Chi Lee** [2,†], **Hsin-Tung Liu** [2] and **Ping-Shan Lai** [3,4,*]

1 Department of Isotope Application Research, National Atomic Research Institute, Taoyuan 325, Taiwan; clpeng@nari.org.tw
2 xTrans Creative Inc., 7F., No. 52, Sec. 3, Nangang Rd., Nangang Dist., Taipei City 115, Taiwan; nausicaa_lee@xt-creative.com (P.-C.L.); ruby_liu@xt-creative.com (H.-T.L.)
3 Department of Chemistry, National Chung Hsing University, No. 145, Xingda Road, Taichung 402, Taiwan
4 Ph.D. Program in Tissue Engineering and Regenerative Medicine, National Chung Hsing University, No. 145, Xingda Road, Taichung 402, Taiwan
* Correspondence: pslai@email.nchu.edu.tw
† These authors contributed equally to this work.

**Abstract:** This study aimed to evaluate Zn/Ag-electrode-printed patches for the transdermal delivery of small molecules through iontophoresis. The Zn/Ag-electrode-printed patches interact with biological liquid electrolytes and generate suitable microcurrents for the iontophoretic delivery of small molecules across the skin. In fluorescein permeation studies, Zn/Ag-electrode-printed patches increased the transdermal depth of fluorescein into the dermis, while the permeation of fluorescein was limited when Zn/C-electrode-printed patches were tested. Further permeation experiments were conducted with 3D skin models, which showed a similar trend to the above, indicating that Zn/Ag-electrode-printed patches had a higher penetration rate compared to the blank. Studies using acetyl hexapeptide-8 as a peptide drug model and sodium ascorbyl phosphate (SAP) as a hydrophilic derivative of ascorbic acid showed that the iontophoretic patch with Zn/Ag electrodes promoted more penetration of drugs than unprinted patches. The permeation of SAP exhibited a two-phase profile with a relatively rapid permeation followed by a sustained, slower permeation. The permeation of acetyl hexapeptide-8 was slower due to its higher molecular weight, but the iontophoretic patch increased the permeation up to 1.5 times more than the unprinted patch. The microcurrent generated by the patch drives the transport of small molecule components through the skin, for the controlled and efficient delivery of therapeutic agents. The flexible design, efficient microcurrent generation, and stable electrodes make the Zn/Ag-electrode-printed patch a promising tool for transdermal drug delivery.

**Keywords:** transdermal delivery; iontophoresis; Zn/Ag electrodes





## 1. Introduction

A generic sheet patch is composed of 80–90% water, making it difficult for the useful ingredients to permeate the skin. Therefore, iontophoresis and electroporation are often used to enhance transdermal drug delivery [1]. In iontophoresis, a low-level electrical current drives charged drugs or other active ingredients into the skin, while electroporation uses brief electrical pulses to temporarily create pathways through the skin for drug delivery. These techniques can overcome the barrier posed by the skin's outer layer and increase skin permeability to deliver active ingredients from the patch to the underlying tissues. Iontophoresis has been used for decades for transdermal drug delivery, including small-molecule drugs, amino acids, peptides, and proteins [2–4]. Iontophoresis has gained renewed interest in recent years due to the increasing number of peptide drugs being developed for various therapeutic applications [5]. The delivery of small charged peptides is difficult using conventional methods, but iontophoresis is an effective solution for

overcoming these challenges [6]. The transdermal delivery of peptides using iontophoresis can improve drug efficacy and patient compliance, reduce the need for frequent injections, and minimize the risk of side effects associated with other routes of administration.

Many power-generating patches require additional equipment or batteries to operate. In our "power generation printing technology", the positive and negative electrodes are reduced to a micrometer size and printed onto the patches, like a battery printed directly on the patch that can self-generate electrical power. Silver (Ag) and zinc (Zn) have antimicrobial properties [7] and are biocompatible [8,9], making them suitable for use in biomedical applications such as iontophoretic patches. Patches with Zn/Ag electrodes can function in biologically relevant electrolyte environments and provide flexibility to conform to curved surfaces of the human anatomy without limiting the current flow for biomedical applications [10]. Therefore, Zn/Ag electrodes were chosen to prepare the iontophoretic patches.

Antioxidants, such as ascorbic acid and sodium ascorbyl phosphate (SAP), are commonly used in skincare products for their antiaging benefits. Ascorbic acid, in particular, has a wide range of benefits for the skin, including antioxidant properties, anti-inflammatory effects, photoprotection, and the stimulation of collagen synthesis. Thus, ascorbic acid is a popular ingredient in antiaging products to treat wrinkles and rejuvenate skin [11]. Acetyl hexapeptide-8 is a synthetic antiaging peptide that inhibits the release of neurotransmitters by interfering with the formation and stabilization of protein complexes required for acetylcholine release from docking vesicles [12]. The inhibition of neurotransmitter release reduces the repetitive contractions of intrinsic muscles during facial expression leading to reduced hyperkinetic facial lines or expression wrinkles [13]. Thus, acetyl hexapeptide-8 is another popular ingredient in antiaging products.

Herein, we designed and manufactured Zn/Ag-electrode-printed patches with electricity-generating printing technology. The Zn/Ag-electrode-printed patches generate electricity for iontophoretic transdermal delivery without an additional power supply (Figure 1). Iontophoresis is the application of an electrical current to drive drug molecules across the skin's epidermis. It operates through mechanisms like electrorepulsion and electro-osmosis, which enhance transdermal drug delivery [14]. Electrorepulsion propels charged molecules into the skin via repulsive forces between like charges, while electro-osmosis facilitates the flow of water, aiding the penetration of active ingredients such as drugs and cosmetics. This technique holds promise for delivering peptide and protein therapeutics by effectively transporting high-molecular-weight compounds across the skin barrier. Utilizing a low-level electric current, iontophoresis enables the delivery of various drugs, including insulin, fentanyl, and lidocaine, as well as other peptides and proteins [15–17].

In this study, we first examined the microstructure and electrochemical characteristic of the Zn/Ag-electrode-printed iontophoretic patch using scanning electron microscopy (SEM) and a potentiostat/galvanostat instrument, respectively. Then, we measured the iontophoretic delivery of fluorescein (NaFI), adopted as a small molecular drug model, into the skin using both SD rat skin and reconstructed human epidermis (RhE) models. The in vitro permeation studies were performed using non-animal artificial film Strat-M® membranes in a modified Franz diffusion cell apparatus. The Franz diffusion cell apparatus was used to measure the transdermal permeation of acetyl hexapeptide-8, a peptide drug model, and SAP, a hydrophilic derivative of ascorbic acid, mediated by the iontophoretic patch.

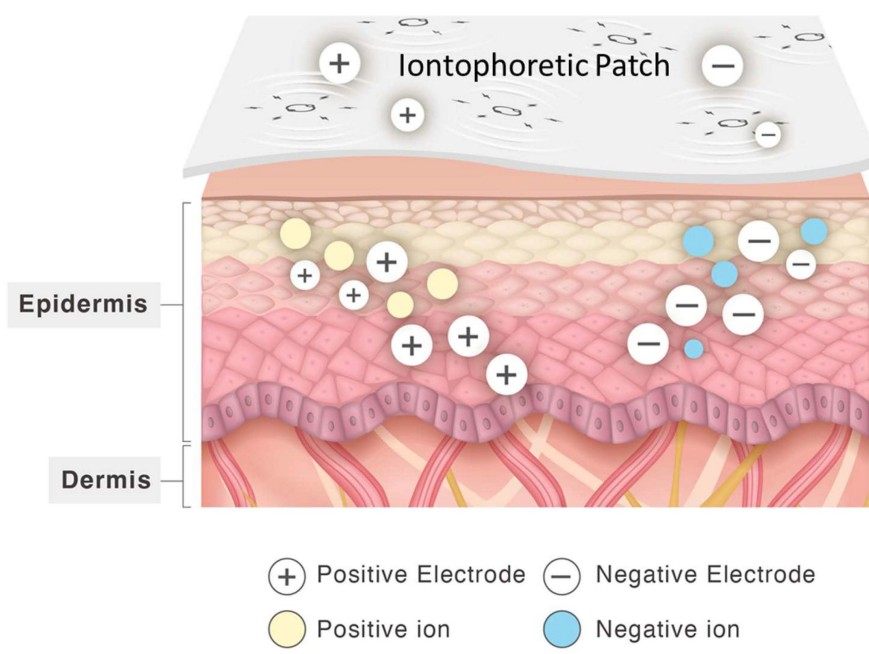

**Figure 1.** Schematic of transdermal delivery by using iontophoretic patch printed with Zn/Ag electrodes.

## 2. Materials and Methods

### 2.1. General

Fluorescein sodium salt (NaFl), SAP, acetyl hexapeptide-8, and phosphate-buffered saline (PBS) powder were purchased from Sigma-Aldrich (St Louis, MO, USA). Silver (Ag) ink (Creative Materials Inc., Ayer, MA, USA) or carbon (C) ink (Creative Materials Inc., Ayer, MA, USA) were used to print the cathode, and an in-house zinc ink, consisting of Zn particles (<10 μm; Sigma Aldrich, St. Louis, MO, USA), was used for the anode. Zn/Ag ink and Zn/C ink were printed on a silk substrate to prepare the Zn/Ag or the Zn/C-electrode-printed patches, manufactured by xTrans Creative Inc (Taipei, Taiwan) [18]. Polyvinyl alcohol (PVA; Sigma Aldrich), a biocompatible binder, was used to hold the Zn particles in the ink and effectively adhere the particles to the supporting fabric during the printing and curing [19–21].

The microstructure of the iontophoretic patch was examined using SEM (JSE-6500F, Jeol, Tokyo, Japan). The reconstructed human epidermis (RhE) models EpiDerm™ (MatTek, Ashland, OR, USA) were obtained for further permeation of fluorescein on 3D Skin.

Electrochemical measurements of iontophoretic patch were performed with the Autolab PGSTAT204 potentiostat/galvanostat instrument (Metrohm, Utrecht, The Netherland). A non-animal artificial film Strat-M® membrane (SKBM02560, Merck, Darmstadt, Germany) was used for transdermal diffusion testing (in vitro permeation studies). The concentrations of permeated substances were measured by a Hitachi high-performance liquid chromatography (HPLC) system (Hitachi L-2000 HPLC, Chiyoda, Japan) coupled to an autosampler (Hitachi L-2200, Chiyoda, Japan), quaternary pump (Hitachi L-2130, Chiyoda, Japan), and diode array detector (Hitachi L-2455, Chiyoda, Japan). Data were acquired with Hitachi software (D-2000 Elite version 1.1). Different types of LC columns were used for development, including the Luna C18 (250 × 4.6 mm, 5 μm, 100 Å; Phenomenex, Torrance, CA, USA) and Luna C18 (150 × 4.6 mm, 5 μm, 100 Å; Phenomenex, Torrance, CA, USA) columns. The mobile phase consisted of 0.01 M $NaH_2PO_4$/Water (pH = 3) as solvent A and methanol as solvent B. Analytes were eluted at a flow rate of 0.5 mL/min using an isocratic elution step (95:5 *v/v*), with a total run time and injection volume of 10 min and 50 μL, respectively.

### 2.2. Microstructure of Iontophoretic Patch

The microstructure of the iontophoretic patch printed with Zn/Ag electrodes was examined with SEM. After the critical drying point, the iontophoretic patch was placed on a conductive substrate, sputter-coated with gold, and viewed with emission SEM.

### 2.3. Electrochemical Measurements of Iontophoretic Patch

Electrochemical measurements were performed with the Autolab PGSTAT204 potentiostat/galvanostat instrument (Metrohm, Utrecht, The Netherland). The measurements were conducted using an iontophoretic patch with a geometric area of 4 cm$^2$ which was immersed in a 0.01 M phosphate-buffered saline aqueous solution (p3813-10PAK, Sigma-Aldrich, USA). During the measurement process, two electrodes were positioned 1 cm apart, with one electrode pinpointed on the printed area and the other on the non-printed area. The measurements were repeated six times, with a detection time of 10 min for each repetition.

### 2.4. Permeation of Fluorescein on Rat Skin

Sprague Dawley (SD) rats were purchased from the National Laboratory Animal Center (Taipei, Taiwan). All animal procedures were performed in accordance with recommendations for the proper use and care of laboratory animals following protocol approval (Approval Number: 111-023R). Skin samples were collected from sacrificed SD rats to measure the permeation of fluorescein (NaFI). The hair was removed and the skin was cleaned with deionized water.

NaFI permeation was optically measured in cross-sectioned skin samples using fluorescence microscopy. The NaFI solution (0.2% *w/v*, 1 mL) was applied to rat skin to compare the transdermal distribution of fluorescein by passive permeation and enhanced permeation with the iontophoretic patch. Iontophoretic patches were attached to rat skin sections. After 5 and 20 min, the iontophoretic patches were detached and the skin was cleansed with deionized water to eliminate residual signal. The skin specimens were placed into OCT solution and vertically sectioned (10 μm-thick) with a cryostat (Leica CM3050S, Leica Microsystems Nussloch GmbH, Heidelberg, Germany). Sections were imaged using a fluorescence microscope (IX71, Olympus, Tokyo, Japan) with an FITC filter (Excitation filter: HQ480, Emission filter: HQ535), and images were captured with a camera (Olympus U-CMAD3) and processed using the Picture Frame™ Application 2.2 software (Optronics, Goleta, CA, USA) [22].

### 2.5. Permeation of Fluorescein on 3D Skin

The patented reconstructed human epidermis (RhE) models EpiDerm™ were used for further permeation test. The EpiDerm™ tissue serves as an analogue to human skin consisted of fully differentiated basal, spinous, granular, and stratum corneum layers.

The 3D skin membranes were incubated at $37 \pm 1$ °C, $5 \pm 1\%$ CO$_2$, and $90 \pm 10\%$ RH before experiment. The activated membranes were then suspended on special cell culture inserts of standard 6-well plates with 0.2 mL of 0.2% NaFI (Sigma-Aldrich, MO, USA) solution applied. Zn/Ag-electrode-printed iontophoretic patches (0.5 cm$^2$) were then suspended on the liquid surface for 12 min as our experimental group. After 12 min, we removed NaFI solution and rinsed with PBS solution for 5 times. We excised the tissues from the inserts and fixed in 10% neutral buffered formalin for 15 min at room temperature. We counterstained cryostat sections with 1 μg/mL DAPI for nuclei staining and recorded data on confocal laser scanning microscope (Olympus FV3000, Tokyo, Japan) at 40× magnification. Two excitation wavelengths were utilized, 405 nm and 488 nm for nucleus (DAPI) and NaFI detection, respectively. Final data calculations were shown as follows:

$$\text{NaFI penetration \%} = \frac{100 \times \text{NaFI penetration depth (μm)}}{\text{Total skin depth (μm)}}$$

$$\text{Relative penetration} = \frac{\text{NaFI penetration \% (experimental group)}}{\text{NaFI penetration \% (control group)}}$$

### 2.6. In Vitro Permeation Studies

Non-animal artificial film Strat-M® membranes and a modified Franz diffusion cell apparatus (Logan DHC-6T, Somerset, NJ, USA) were used to compare changes in skin permeation due to the Zn/Ag-electrode-printed patches with unprinted patches [23]. The adjacent surface and receptor compartment were set up as described in previous studies [24]. The temperature of the artificial membrane surface and receptor compartment was maintained at 32 °C $\pm$ 5 °C. The artificial membranes were soaked in PBS for 30 min and mounted onto the Franz cell apparatus for an additional 30 min for equilibration and hydration. To determine if the iontophoretic patch enhanced permeation, experiments were divided into unprinted patch (blank) and iontophoretic patch groups. In the iontophoretic patch experiments, Zn/Ag-electrode-printed iontophoretic patches (1 cm$^2$) were applied to the artificial membrane. One ml of 4% SAP or 500 ppm acetyl hexapeptide-8 solutions were applied to the surface of the iontophoretic patch adjacent to the artificial membrane. At 10, 20, 30, 60, and 120 min, all samples were collected from the receptor compartment reservoir and stored at −20 °C. The subtracted volume was replaced with fresh phosphate-buffered solution. The concentrations of SAP and acetyl hexapeptide-8 were measured by HPLC system. The SAP and acetyl hexapeptide-8 were detected at 252 and 225 nm, respectively [25]. Each permeation experiment was conducted in triplicate. For the calibration curves of SAP and acetyl hexapeptide-8, the concentration ranges were 0.24 to 250 ppm and 0.195 to 100 ppm, respectively. The R-squared (R$^2$) values for the calibration curves of SAP and acetyl hexapeptide-8 were both above approximately 0.999.

### 2.7. Statistical Analysis

The data from this study were presented as mean $\pm$ standard deviation, and the *t*-test was used to test for significant difference between groups. Values of $p < 0.005$ or $p < 0.001$ were considered statistically significant.

## 3. Results and Discussion

### 3.1. Microstructure of Iontophoretic Patch (Microstructure and Porosity)

Figure 2 shows the SEM images of the patches printed with Zn/Ag electrodes or Zn electrodes. The images show the Zn/Ag particles covering the silk fibers (Figure 2A,B). The Zn electrodes exhibited a spherical morphology, while the Ag electrodes were flake-shaped. The Zn particles did not fully cover the underlying silk fibers (Figure 2C,D). Furthermore, the SEM images of the Zn/Ag printed pattern vividly demonstrated that the metallic particles formed a layer over the underlying silk fibers, and the energy-dispersive X-ray spectroscopy (EDX) analysis, as outlined in our previous study [18], confirmed the presence of Ag and Zn on the Zn/Ag electrode-printed patches. SEM should be used to characterize the microstructure of the iontophoretic patches, as this information provides insight into the electrochemical properties of the electrodes and the overall performance of the patches.

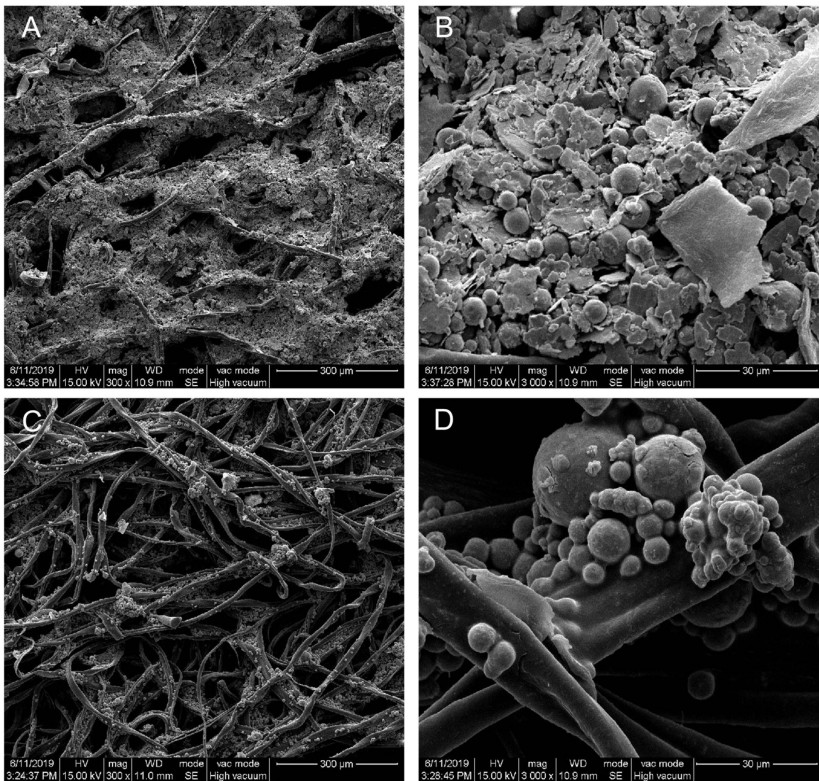

**Figure 2.** Scanning electron microscope (SEM) images of patches printed with Zn/Ag electrodes (**A,B**) and Zn/C electrodes (**C,D**) with the underlying silk fibers. Magnification: (**A,C**) 300×; (**B,D**) 3000×. The scale bars are 300 and 30 μm at magnifications of 300× and 3000×, respectively.

### 3.2. Electrochemical Measurements of Zn/Ag-Electrode-Printed Patch

Before we examined the effectiveness of the Zn/Ag-electrode-printed iontophoretic patch, we first characterized the performance of the patch using a potentiostat/galvanostat instrument. The polarization and current curves were shown in Figure 3. A 0.01 M PBS aqueous solution was used as the electrolyte. The voltage and current measurements provide valuable information regarding the performance of the Zn/Ag-electrode-printed iontophoretic patch. The voltage curve showed an initial increase from 0.6 V to 0.74 V within 30 s, followed by stabilization for the remainder of the 10 min detection period. Similarly, the current measurements showed a gradual decrease from 0.04 mA to 0.02 mA, before stabilizing at 0.03 mA. These patterns suggest that the patch is capable of maintaining a consistent voltage and current output, which is a crucial factor for efficient drug delivery.

Overall, these results demonstrate that the Zn/Ag-electrode-printed iontophoretic patch has a consistent and stable performance in terms of voltage and current output, indicating its potential for use as a reliable and effective non-invasive drug delivery system.

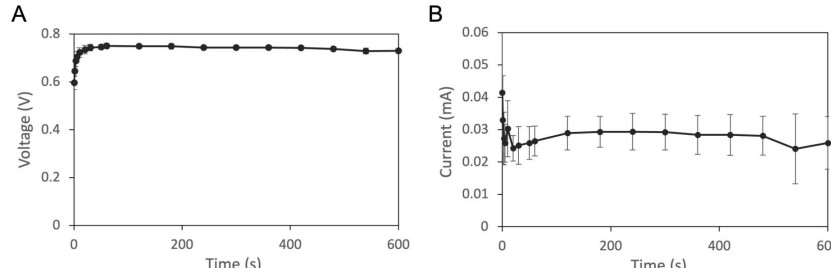

**Figure 3.** Electrochemical measurements of Zn/Ag-electrode-printed patch. Polarization (**A**) and current (**B**) curves. Experiments were repeated in triplicate and the data were presented as mean ± standard deviation (*n* = 3).

### 3.3. Permeation of Fluorescein on Rat Skin

To examine the effectiveness of the microcurrent produced by the Zn/Ag-electrode-printed patch, we examined the iontophoretic delivery of fluorescein (NaFI). The permeation of NaFI was optically measured from cross-sectioned skin samples. NaFI solution was applied to rat skins to compare the transdermal permeation of fluorescein by passive permeation and enhanced permeation with the iontophoretic patch.

The Zn/Ag-electrode-printed patches were effective in increasing the transdermal depth of fluorescein, as shown in Figure 4. After 5 min of iontophoretic delivery using the Zn/Ag-electrode-printed patches (Figure 4B), NaFI penetrated to a depth of approximately 0.3 mm into the skin. After 20 min of delivery, the depth of NaFI penetration increased to around 0.4 mm (Figure 4C). In contrast, the control experiment using blank patches and the Zn/C-electrode-printed group showed limited fluorescent permeation, which was concentrated in the stratum corneum layer (Figure 4A,D,E). These findings indicate that the use of biocompatible Zn/Ag-electrode-printed patches is an effective method for enhancing the transdermal delivery of drugs and other small molecules. The development of biocompatible, self-generating microcurrent devices has great potential for improving the delivery of drugs and other small molecules for a range of biomedical applications, including the treatment of skin conditions and wound healing [18]. The ability to self-generate microcurrents in the presence of biological fluids increases the versatility and usefulness of the patches in a range of clinical settings. In addition, eliminating the need for an external power supply reduces the complexity and cost of the device, making it more accessible to patients.

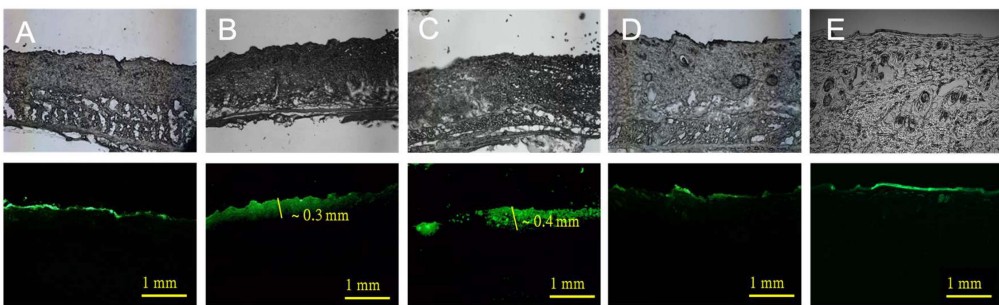

**Figure 4.** Permeation of NaFI solution topically applied with or without electrode-printed patch on skin. All images in this figure are magnified at 200. Control (**A**), using no electrode patch; Zn/Ag 50, iontophoretic delivery by Zn/Ag-electrode-printed patch for 5 min (**B**) and 20 min (**C**), respectively; and Zn/C 50, iontophoretic delivery by Zn/C-electrode-printed patch for 5 min (**D**) and 20 min (**E**), respectively (upper column, differential interference contrast (DIC) images; and bottom column, fluorescence images).

### 3.4. Permeation of Fluorescein on 3D Skin

The in vitro permeation was established using reconstructed human epidermis (RhE) models EpiDerm™ for evaluating whether adding the Zn/Ag-electrode-printed iontophoretic patch would increase the permeation of fluorescein (NaFI). The permeation depths were measured and compared between the NaFI-only control group (n = 6) and the NaFI with Zn/Ag-electrode-printed iontophoretic patches experimental group (n = 9). After 12 min of fluorescein treatment, the permeability was calculated as the ratio of the NaFI depth versus the total epidermis depth, as shown in Figure 5. In the control group, the average permeation depth of NaFI was 8.8 μm, which corresponds to a permeability of approximately 10% with a total epidermis depth of 91.5 μm. In contrast, the permeation depth of NaFI in the experimental group was 24.2 μm and the epidermis thickness was 118.5 μm. This represents a penetration rate of 20.7% and a statistically significant difference ($p < 0.005$) between the two groups. In Figure 5, the calculation is displayed as the ratio of the NaFI depth versus the total stratum corneum (SC) thickness. The SC thickness in the control group and the iontophoretic group was 50.4 μm and 54.9 μm,

respectively. Corresponding to the permeability of 17.78% and 46.8%, with the statistically significant p value of $p < 0.001$. The results showed a higher transdermal rate from the aid of the Zn/Ag-electrode-printed patch as compared to the fluorescein-only group. The relative penetration was 2.07 times and 2.63 times higher in comparison with the epidermis and SC, respectively (Figure 5).

In conclusion, the use of the Zn/Ag-electrode-printed iontophoretic patch in combination with fluorescein has been shown to significantly increase the permeation of the substance through the reconstructed human epidermis models EpiDerm™. The permeation depth of NaFI in the experimental group was found to be 2.07 times higher in comparison to the control group when measured against the total epidermis depth and 2.63 times higher when measured against the total SC thickness (Figure 5E). These findings suggest that the use of the Zn/Ag-electrode-printed iontophoretic patch has the potential to improve transdermal drug delivery and provide an alternative route for drug administration. Further studies can be carried out to explore the efficiency of the patch in delivering other substances and to optimize its design for a better performance.

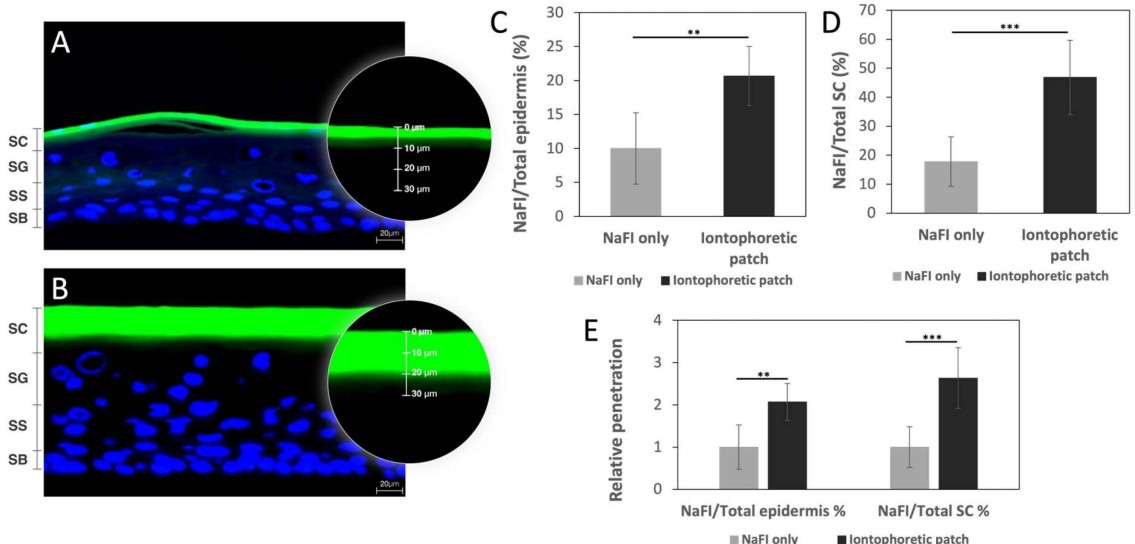

**Figure 5.** The EpiDerm™ confocal imaging and permeation study analyzed the effects of NaFI solution topically applied with and without an iontophoretic patch. Image of NaFI solution (green) topically applied in control group (**A**) and experimental group (**B**). Comparison of the permeation depth of NaFI solution in relation to the total epidermal thickness (**C**) and the SC thickness (**D**) in the control and experimental groups. Relative penetration was also assessed in comparison to the epidermis or SC (**E**). Data represent mean ± SD ($n_{control}$ = 6; $n_{experimental}$ = 9); the results showed a significant difference in iontophoretic group against NaFI-only group in the epidermis (** $p < 0.005$) and SC (*** $p < 0.001$). Counterstaining of cell nuclei with DAPI (blue). Scale bar: 20 μm. Stratum corneum (SC); stratum granulosum (SG); stratum spinosum (SS); stratum basale (SB).

### 3.5. In Vitro Permeation Studies

The permeation of sodium ascorbyl phosphate (SAP) and acetyl hexapeptide-8 was established using an artificial film Strat-M® membrane and a Franz diffusion cell assay. The cumulative permeated amount for the microcurrent-electrode-printed patches was compared with the unprinted patches at 10, 20, 30, 60, and 120 min. Permeability was calculated as the ratio of the permeation amount caused by the microcurrent-electrode-printed patch divided by the permeation amount using the unprinted patch.

Figure 6A shows the in vitro permeation of SAP through the artificial film Strat-M® membrane mediated by the iontophoretic patch (microcurrent-electrode-printed patch). The in vitro permeation of SAP had a two-phase permeation profile: a relatively rapid permeation of SAP before 30 min, followed by a sustained, slower permeation. In a short

time (<30 min), the microcurrent-electrode-printed patch promoted the permeation of SAP, and the penetration rate gradually slowed down after 30 min. The permeation ratio of the iontophoretic patch to the unprinted patch gradually decreased from 3 times at 10 min to about 1 time at 120 min.

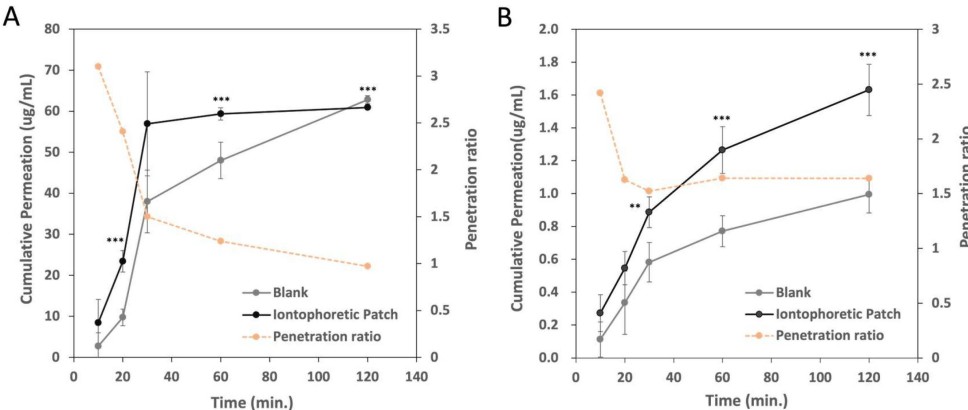

**Figure 6.** In vitro penetration of (**A**) SAP (sodium ascorbyl phosphate) and (**B**) acetyl hexapeptide-8 through artificial film Strat-M® membrane by unprinted patch (blank) or microcurrent-electrode-printed patch for 2 h. Experiments were performed in triplicate and data represent mean ± SD (n = 3). The results showed a significant difference between blank and iontophoretic group (** $p < 0.005$, *** $p < 0.001$).

We also compared the transdermal efficiency of acetyl hexapeptide-8, as a peptide drug model, by passive permeation (unprinted patch) and permeation mediated by the iontophoretic patch using the Franz diffusion cell apparatus, as shown in Figure 6B. The microcurrent-electrode-printed patch promoted the penetration of acetyl hexapeptide-8. The permeation ratio between the iontophoretic patch and the unprinted patch decreased from 2.5 times at 10 min to 1.5 times at 30 min. The penetration rate was maintained at about 1.5 times after 30 min. This result indicates that the iontophoretic patch accelerated the transdermal permeation of acetyl hexapeptide-8 about 1.5 times more than the unprinted patch (passive permeation). Acetyl hexapeptide-8 had a slower permeation rate than SAP, possibly due to its higher molecular weight or the properties of zwitterion. However, the iontophoretic patch sustained the permeation of acetyl hexapeptide-8. Despite the relatively high molecular weight (>800 g/mol) of the peptides used in this study, transdermal delivery was significantly enhanced. This sustained permeation suggests that the iontophoretic patch represents a viable method for the transdermal delivery of the peptide, maximizing the exposure of the active ingredients in the skin tissue.

Iontophoresis refers to the application of a physiologically acceptable electrical current to drive drug molecules across the epidermis of the skin [2–4]. The main mechanisms of iontophoresis to enhance transdermal drug delivery are electrorepulsion and electro-osmosis. Electrorepulsion transports charged molecules into the skin by the repulsive force of the same charge between the charged molecule and the electrodes. Electro-osmosis is the net flow of water from the anode to the cathode under the influence of an electric current, resulting in the penetration of active ingredients, such as drugs and cosmetics, into the skin [14].

Iontophoresis, which can effectively transport high-molecular-weight compounds across the skin barrier, is a promising technique for delivering peptide and protein therapeutics. Iontophoresis involves the use of a low-level electric current to facilitate the delivery of charged molecules through the skin. This technique delivers a variety of drugs, including insulin, fentanyl, lidocaine, and other peptides and proteins [15–17]. Compared to traditional oral delivery, iontophoresis offers several advantages. Iontophoresis can directly deliver drugs to their target sites while avoiding first-pass metabolism in the liver and degradation in the stomach and intestine. Thus, iontophoresis can improve bioavailability and therapeutic efficacy. In addition, iontophoresis is noninvasive and painless, which is especially important for patients who require frequent drug administration. Therefore,

iontophoresis has the potential to deliver both small- and high-molecular-weight drugs stably, making it a promising method for transdermal drug delivery.

## 4. Conclusions

In conclusion, the results of this study demonstrated that the Zn/Ag-electrode-printed patches are a feasible method for the transdermal delivery of small-molecule components through iontophoresis. The patch design facilitates the controlled and efficient delivery of therapeutic agents. The Zn/Ag electrodes are stable and have a good electrical conductivity, making them well-suited for iontophoretic patches. The permeation studies using NaFl, SAP, and acetyl hexapeptide-8 demonstrate that the Zn/Ag-electrode-printed patches increase the transdermal depth of the permeated components compared to unprinted patches. These results suggest that the Zn/Ag-electrode-printed patches are safe and effective for transdermal drug delivery via their self-generated microcurrent.

**Author Contributions:** C.-L.P. performed the experiments; analyzed the data; and wrote the paper. H.-T.L. designed and performed most experiments and contributed to data analysis. P.-C.L. and P.-S.L. jointly participated in the project initiation, experimental design, and data analysis. All authors have read and agreed to the published version of the manuscript.

**Funding:** This research received no external funding. The APC was funded by Pei-Chi Lee (CEO at xTrans Creative Inc., 7F., No. 52, Sec. 3, Nangang Rd., Nangang Dist., Taipei City 115, Taiwan).

**Institutional Review Board Statement:** All procedures were performed following approved protocols that were developed in accordance with recommendations for the proper use and care of laboratory animals (approved number: 111-023R).

**Informed Consent Statement:** Not applicable.

**Data Availability Statement:** The data presented in this study are contained within the article.

**Conflicts of Interest:** Pei-Chi Lee and Hsin-Tung Liu are employed by xTrans Creative Inc., Taiwan. The authors declare that no competing interests exist.

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
