# Peer review of "Advancing Transdermal Delivery by Zn/Ag-Electrode-Printed Iontophoretic Patch with Self-Generating Microcurrents"

_scipharm, doi:10.3390/scipharm92020026_

Round 1

Reviewer 1 Report

Comments and Suggestions for Authors

Iontophoresis non-invasively, bypassing first-pass metabolism, allows the transdermal delivery of small and large API molecules that would not usually permeate the skin. The authors in this paper investigated the effectiveness of the obtained iontophoretic patches with Zn/Ag electrodes in the transdermal delivery of sodium ascorbyl phosphate and ascorbic acid.

The work addresses an interesting issue, indicating the possibility of a future application of Zn/Ag electrodes-printed patches in iontophoresis.

Comments to authors:

- Lines 99-110; the information in this paragraph regarding the equipment, reagents used in the experiment should be found in the subsections where they were used for the study.

- Figure 5 not very legible

- No information is available on what statistical methods were used to compare the figures obtained.

- Please indicate the number of Zn/Ag Electrodes-Printed Iontophoretic Patches obtained.

- For each experiment described, there should be an indication of how many times it has been repeated.

Reviewer 2 Report

Comments and Suggestions for Authors

Dear Authors

Let me, first congratulate you for a such interesting work. The study of this iontophoretic mechanism seems very promising to improve the transport of molecules into the skin.

In my opinion, you should include a picture showing the different components of the system and defining on it the characteristics that you modifiy during the experiments. The changes of rat skin to artificial skin, should be better explained, because the chemical character of the artificial layers would be very important when explaining the improvement due to the voltage or, simply due to the chemical affinity. Please, consdier to add the chemical character of the memebranes and explain in more detail.

Many thanks

Reviewer 3 Report

Comments and Suggestions for Authors

Advancing Transdermal Delivery by Zn/Ag Electrodes-Printed Iontophoretic Patch with Self-Generating Microcurrents

This study aimed to evaluate Zn/Ag electrodes-printed patches for transdermal delivery of small molecules through iontophoresis. In fluorescein permeation studies, Zn/Ag electrodes-printed patches increased the transdermal depth of fluorescein into the dermis, while permeation of fluorescein was limited when Zn/C electrodes-printed patches were tested. Further permeation experiments were conducted with 3D skin models, which showed a similar trend to the above, indicating that Zn/Ag electrodes-printed patches had a higher penetration rate compared to the blank. Studies using acetyl hexapeptide-8 as a peptide drug model and sodium ascorbyl phosphate (SAP) as a hydrophilic derivative of ascorbic acid showed that the iontophoretic patch with Zn/Ag electrodes promoted more penetration of drugs than unprinted patches. This research holds significant value for advancing the development of innovative materials for delivery of small molecules across the skin and aligns with the scope of the Scientia Pharmaceutica Journal. I recommend acceptance after a revision that addresses the following questions:

- Could you provide a more detailed explanation of how the inclusion of APIs within the Zn/Ag electrodes-printed patches influenced the penetration of drugs across the skin, and what factors contributed to this outcome? Is special preparation of the skin surface needed to create a microcurrent?

 - Please provide a more comprehensive discussion regarding the little difference found in the penetration of fluorescein into rat skin after 5 minutes and after 20 minutes.

- Can you elaborate on the distinctions observed in the in vitro penetration profiles between printed and unprinted patches?

- How can the findings of this study be translated into practical implications for the pharmaceutical industry and the processes involved in drug formulation? What specific benefits can be expected?

- Were there any unanticipated results or challenges encountered during the course of the study? If so, how were these challenges addressed, and what insights were gained from them?

- Do you consider carrying on stability studies to complete this work?

- Please include the calibration curve used and the concentration range for HPLC method validation for the two drugs.

- Figures: kindly use a large font while making images.  

- Figure 1: improve the representation of the iontophoretic printed patch.

- Please explain better the SEM method.

- In the Permeation of Fluorescein on Rat Skin Methos, kindly indicate which part of the rat skin was used.

- Line 196, please correct this typo: Figure3.

Round 2

Reviewer 1 Report

Comments and Suggestions for Authors

The authors made corrections to the manuscript as suggested.